# The Bright Future of mRNA as a Therapeutic Molecule

**DOI:** 10.3390/genes16040376

**Published:** 2025-03-26

**Authors:** Dora Emma Vélez, Blanca Licia Torres, Greco Hernández

**Affiliations:** 1mRNA and Cancer Laboratory, Unit of Biomedical Research on Cancer, Instituo Nacional de Cancerología (National Institute of Cancer, INCan), Mexico City 14080, Mexico; doravelezuriza@gmail.com (D.E.V.); blanca.torres@comunidad.unam.mx (B.L.T.); 2Escuela de Medicina y Ciencias de la Salud, Tecnológico de Monterrey, Mexico City 14380, Mexico

**Keywords:** mRNA, cancer, delivery systems, infectious diseases, therapeutic mRNA

## Abstract

The rapid success of messenger (m) RNA vaccines against COVID-19 has pushed the mRNA to the forefront of drug research. The promise of mRNA-based therapeutics and vaccines in other areas is not new but is now emerging stronger. We review basic concepts, key historical aspects, and recent research on mRNA as a therapeutic molecule to fight infectious diseases and cancer. We also show a current patent perspective of this field. Altogether, we describe that the technology of mRNA as a therapeutic molecule is a rapidly moving field aiming for a bright future.

## 1. Introduction

Back in 1990, Wolff and collaborators proposed that the intracellular expression of messenger RNA (mRNA) or DNA-encoded antigens could be an alternative approach to vaccine development [1]. Three years later, Martinon and colleagues explored mRNA–liposome complexes to induce an immune response against the influenza virus in mice, stimulating the production of anti-influenza cytotoxic T lymphocytes (CTLs) in vivo [2]. This study marked the beginning of the long road in the development of mRNA vaccines. During the 1990s, some mRNA was used in preclinical approaches for vaccination. However, it was not until 2020 during the COVID-19 pandemic that the technological innovations achieved in past decades paved the road for the first Food and Drug Administration (FDA)-approved mRNA vaccine, a highly effective vaccine against the SARS-CoV-2 virus. This event marked a milestone in the mRNA vaccine field, strongly spurring interest in them for the treatment of other maladies, including cancer and infectious diseases. Thus, mRNA as a therapeutic molecule has rapidly emerged as a promising field in biomedical research. Indeed, the market of therapeutic mRNA was estimated at 20.83 billion USD in 2025 and is expected to reach 42.64 billion USD by 2034 [3].

mRNA vaccines have proved their ability to induce antigen-specific humoral and cellular immunity. The best vaccines stimulate helper T cell responses to assist in antibody production and induce cytotoxic CD8+ T cells [4]. Over the last decades, several nucleic acid-based vaccine strategies have been developed, but mRNA has several advantages against other platforms: (1) mRNA vaccines are safer because they do not integrate into the cellular genome; (2) they induce a stronger immune response against pathogens or damaged cells; (3) mRNA accesses the translation machinery faster and more easily, making antigen production a quick process; (4) multiple antigens can be simultaneously delivered and can stimulate the cross-presentation of several epitopes in antigen-presenting cells (APCs); and (5) they have a lower cost and their development is faster than other platforms [5,6,7,8,9]. We review the critical milestones in the history of mRNA research that led to the development of mRNA vaccines. We also describe the state of the art of this rapidly moving field regarding infectious diseases and cancer.

## 2. mRNA, the Promising Molecule

mRNA was discovered in 1961 [10,11]. It is a single-stranded molecule that carries genetic information from DNA to ribosomes to synthesize proteins. In Figure 1, we depict the features of a typical eukaryotic mRNA molecule. At the 5’ end, there is a cap 7-methyl guanosine (m^7^G) linked to the first nucleotide of the mRNA by a unique 5’-5’ triphosphate. This structure increases mRNA stability by protecting against 5’→3’ exonucleolytic degradation, as well as splicing accuracy and efficiency. The cap also participates in intracellular transport and promotes translation by interacting with eukaryotic translation initiation factor (eIF) 4E [12,13]. mRNA possesses untranslated regions (UTRs) at both ends. The 5’-UTR contains regulatory elements involved in translation initiation and stability, and the 3’-UTR is involved in mRNA turnover, translation, and localization [14,15,16]. The mRNA-coding sequence (CDS) or open reading frame (ORF) encodes the protein, most of the time beginning with the start codon AUG and ending with one of the stop codons UAA, UAG, or UGA. Finally, downstream of the 3’-UTR, there is a stretch of 50–250 adenosine residues—the poly(A) tail—that stabilizes the molecule; promotes translation by interacting with the cap trough the interaction of the poly(A)-binding protein (PABP) with the translation initiation factor eIF4G to form a loop-like conformation; protects mRNA from enzymatic degradation in the cytoplasm; and participates in the export of mature mRNAs to the cytoplasm [17]. Because of its single-stranded nature, mRNA folds into complex secondary structures such as hairpin loops, internal loops, and base pair stackings. Altogether, these structures play crucial roles in mRNA turnover, translation, transport, and storage during the development of therapeutic mRNAs, including mRNA vaccines [18,19,20].

## 3. Origins and Innovations to Develop mRNA Vaccines

Over the past three decades, mRNA cancer vaccines have evolved from basic research to clinical applications. In the 1990s, the concept of using mRNA for vaccination purposes was first explored. After Martinon’s findings, Zhou and collaborators developed in 1994 a self-amplifying mRNA vaccine using the Semliki Forest virus, which successfully elicited an immune response in mice [21]. Later, Conry et al. proved the feasibility of using an mRNA encoding tumoral antigens as a cancer vaccine by the immunization of mice with an mRNA encoding the human carcinoembryonic antigen (CEA), which were later challenged with CEA-expressing tumor cells. These mice showed higher antibody levels than those that only received CEA-expressing tumor cells (control group) [22].

Despite these achievements, in the early 2000s, pioneering studies led by Katalin Karikó and Drew Weissman demonstrated that exogenous mRNA can elicit both a robust immune response and inflammation by activating innate immune pathways. The Karikó–Weissman team observed that the transfection of human DCs with exogenous mRNAs, or even with non-coding ribonucleotide homopolymers, provoked inflammatory cytokines [23,24,25]. Upon entry into the body, mRNA is recognized by pattern recognition receptors (PRRs)—specifically, the toll-like receptors (TLRs) expressed on antigen-presenting cells (APCs). This recognition initiates downstream signaling cascades, triggering the secretion of inflammatory cytokines such as TNF-α, IL-6, IL-12, and type I interferons, along with the recruitment of immune cells [23,24,25]. Subsequently, APCs present mRNA-derived antigens, thereby promoting the activation of adaptive immunity and amplifying the inflammatory response as part of the immune surveillance mechanism. Moreover, dendritic cells (DCs) and macrophages utilize RNases to degrade foreign mRNA [26,27,28]. In addition, double-stranded RNA (dsRNA) can induce a halt in protein synthesis by promoting the phosphorylation of eukaryotic translation initiation factor 2 (eIF2) via the activation of eIF2α protein kinase R (PKR) [29]. Although mRNA vaccines are designed to stimulate the immune system, these responses may lead to a life-threatening exacerbated immune response. To address these problems, chemical modifications to the mRNA and the formulation of delivery vehicles have been developed [20,30,31], enabling the reduction of the immunogenicity of synthetic mRNAs while enhancing their stability and translation efficiency for their safer use in vaccines [20,32,33].

### 3.1. mRNA Sequence Engineering

The structure and regulatory elements of mRNAs are critical for gene expression and their use in therapeutic applications like vaccines. Thus, engineering the mRNA sequence can improve its stability and expression [20,30,31,32,34]. These include (1) the addition of a 5’ cap analog; (2) optimizing the 5’- and 3’-UTR sequences, length, and G/C content; (3) optimizing the size of the poly(A) tail; (4) optimizing codon usage in the ORF; and (5) substituting uridine with pseudouridine or other chemically modified nucleotide analogs [20,30,31]. The use of modified nucleotides and optimizing codon usage improves the translation elongation rate by modifying the guanine and cytosine content [35,36]. Stable secondary structures and hairpin loops in some mRNAs should be avoided in the ORF. However, each mRNA should be tested in this respect [9,37]. Excessively stable secondary structures within the ORF could slow down or stall ribosomal progression [9,37], while moderate secondary structures could enhance mRNA stability, indirectly improving translation efficiency [34]. In mRNA therapeutics, the optimal poly(A) tail length should be determined. A tail length between 100 and 150 nucleotides offers a good balance to improve stability and efficiency of translation [38]; chemical modifications such as phosphorothioate can increase translational efficiency [20,26]. Finally, changes in the structure of the poly(A) tail like the addition of 40–60 adenosines separated by a spacer element or the chemical addition of multiple tails enhance translation efficiency [39,40].

DNA is known to activate toll-like receptors (TLRs) (particularly TLR9), which belong to the family of PRRs and play an important role in pathogen recognition; on the other hand, double-stranded RNA (dsRNA) activates TLR3 and induces type I interferon. TLR7 and TLR8 sense single-stranded RNA (ssRNA). Interestingly, transfer RNAs (tRNAs) are enriched in modified nucleosides, making them non-inflammatory. Therefore, Karikó and Weissman generated RNAs with modified nucleotides, such as pseudouridine (Ψ) or 1-methyl pseudouridine (m1Ψ) for uridine or 5-methylcytidine (m5C) for cytidine [23,24,41,42]. These changes rendered the RNA non-immunogenic and, consequently, non-inflammatory, due to a decrease in RNA recognition by TLRs [23,24,41,42]. Indeed, this was a milestone discovery, which later enabled Moderna and BioNTech/Pfizer to develop effective, US Food and Drug Administration (FDA)-approved mRNA vaccines against COVID-19, both based on 1-methylpseudouridine-containing mRNA [23,24,41,42,43]. These ground-breaking discoveries made by the 2023 Nobel laureates Karikó and Weissman, “for their discoveries concerning nucleoside base modifications that enabled the development of effective mRNA vaccines against COVID-19”, triggered the birth of a new era in therapeutic mRNA. They played a pivotal role in the development of mRNA technology that prevented undesirable inflammatory reactions and increased the translation of mRNA-expressed antigens. These findings highlight the need to consider certain trade-offs in mRNA vaccine design. Specifically, it is crucial to decide whether to prioritize greater translation efficiency—leading to a higher quantity of produced peptide at the cost of reducing the adjuvant effect of mRNAs—or to maintain the adjuvant effect by not modifying the nucleotides, resulting in lower translation efficiency [44]. In the case of cancer vaccines, the reduction in toll-like receptor (TLR) activation due to the incorporation of modified nucleotides must be considered, as it may influence the antitumor signals of the innate immune system, thus affecting antigen presentation and cytotoxic T lymphocyte (CTL) differentiation [45]. As a result, both modified and unmodified mRNA vaccines are being developed that are both safe and effective [46,47]. It is noteworthy that in 2022, Sittplangkoon et al. found that substituting 100% uridine with N1-methylpseudouridine (m1ψ) in an mRNA vaccine in a mouse melanoma model promoted tumor growth and metastasis [48]. Consequently, further studies are needed to fully understand the impact of modified nucleotides on carcinogenesis.

### 3.2. mRNA Encapsulation: From Micelles and Liposomes to Lipid Nanoparticles in mRNA Vaccines

The successful delivery of therapeutic mRNA to target cells requires safe, effective, and stable delivery systems for nucleic acid protection from degradation. Thus, scientists have developed viral vector-based and non-viral RNA transport systems to ensure effective RNA delivery that evades RNA decay and increases both the delivery to target cells and immunogenicity. These systems are designed to overcome critical obstacles such as the tumor microenvironment and immune evasion. Concurrent advances in the development of synthetic materials for RNA encapsulation, such as polymers and lipid molecules like LNPs, have triggered research into non-viral-based delivery systems that led to the emergency use authorization and the FDA approval for intramuscularly inoculated LNP-based mRNA SARS-CoV-2 vaccines [49]. In the following, we describe the delivery systems currently being used.

The large size of some mRNAs and their negative charge make them unable to cross the anionic lipid bilayer of cell membranes [50]. Therefore, research has focused on delivery methods including routes of administration, pharmaceutical carriers, and drug packaging to overcome the challenges of mRNA instability, inefficient delivery, and immune activation, paving the way for safe and effective mRNA therapeutics.

Since the first proof-of-concept animal study in 1990 [1], numerous strategies have been explored to overcome the instability and immunogenicity of synthetic mRNAs. Successful intracellular mRNA delivery involves viral and non-viral vehicles. Non-viral vehicles include lipids, lipid-like materials, polymers, and protein derivatives. Lipids are amphiphilic molecules that contain three domains: a polar head group, a hydrophobic tail region, and a linker between the two domains [51]. They can form diverse structures such as lipid nanoparticles (LNPs), micelles, and liposomes, each with unique characteristics and applications for the delivery of chemotherapeutic drugs or nucleic acids [52]. Micelles are small (<30 nm), single-layer lipid nanosized particles with a hydrophobic core and hydrophilic shell used successfully to solubilize poorly water-soluble drugs. On the other hand, liposomes, a term coined in 1960 and whose use in the transport of nucleic acids has been reported since 1976, are extremely versatile carriers, capable of transporting hydrophobic and hydrophilic molecules, such as small molecules, nucleic acids, and proteins. Liposomes are the early version of LNPs, which came in the early 1990s as the first nanomedicine delivery platform for medical purposes [53]. Nowadays, LNPs have been successfully used in the clinic for small molecule delivery [30] and have emerged as leading non-viral carriers for mRNA in clinical applications. The first LNP drug that received regulatory approval from the European Medicines Agency (EMA) in 1990 and from the FDA in 1997 was AmBisome, developed by the company Gilead. The encapsulated drug was Amphotericin B, indicated for fungal infections and leishmaniasis. Subsequently, companies continued encapsulating different drugs, such as Doxorubicin, for some cancers like Kaposi’s sarcoma, ovarian, and breast. However, it was not until 2018 that the earliest LNP–nucleic acid drug was approved by the FDA and the EMA. It was Patisiran (Onpattro^®^), developed by Alnylam Pharmaceuticals as an siRNA–LNP complex designed to reduce transthyretin protein formation and to treat hereditary transthyretin-mediated amyloidosis, representing a milestone in nucleic acid therapeutics [54].

Lipid combinations confer the biological and structural properties essential for LNPs. Thus, current approved LNP formulations by the FDA and EMA include four main lipids: (1) cholesterol; (2) a polyethylene glycol (PEG)–lipid; (3) an ionizable or cationic lipid; and (4) conjugate helper lipids such as dioleoylphosphatidylethanolamine (DOPE). These lipids enhance the LNP stability, promote the formation of monodisperse nanoparticles, aid in the efficient encapsulation of nucleic acid, improve cellular uptake, and stimulate the endosomal escape of nucleic acid cargo [55].

Size matters. In vaccine development, size plays an important role in the in vivo stability, distribution, and immune response provoked by the LNP–mRNA complexes. Nanoparticles (NPs) smaller than 20 nm will be eliminated. In contrast, the 20–200 nm NPs could be engulfed by DCs and drain to lymphatic nodes, increasing vaccine immunogenicity by improving antigen presentation, generating a strong immune response rather than bigger LNPs. NPs larger than 200 nm will remain at the administration site where they are taken up mostly by macrophages and carried to lymph nodes [56,57]. Size can also determine the cellular internalization path. NPs sized 20–200 nm will be endocytosed, and NPs smaller than 100 nm will be taken by clathrin- or caveolae-dependent endocytosis; the largest NPs (>500 nm) are usually phagocytized [58,59]. Furthermore, there is evidence that size affects the type of immune response. For example, NPs larger than 1 μm are capable of initiating a Th1 response and stimulating IL-12 production, while particles smaller than 500 nm are prone to induce Th2 responses throughout IL-1β secretion [60]. Additionally, LNPs stimulate antibody production and B cell-induced immune memory [30]. These aspects also make 20–200 nm the optimal size because it enables nanoparticles to endure fluid flow, such as that of blood when crossing the interstitial space [61]. The efficiency in every LNP trait previously described could be optimized by modifying the core and shell layer compositions [62]. Table 1 summarizes the typical LNP composition and function for mRNA delivery.

### 3.3. Innovative Improvements in the Design of Lipid Nanoparticles

Since the COVID-19 pandemic, the use of LNPs has dominated the mRNA delivery field, to the extent that the use of non-lipid carriers is unusual in the context of mRNA vaccination. The LNP delivery system is key to the success of gene replacement and vaccines, protecting nucleic acid cargo from enzymatic degradation until its delivery to the cytosol of the target cell [55]. LNPs are organ- and cell-specific, delivered by variations in lipid composition during formulation, which changes the nanoparticle size and charge, leading to selective accumulation in different organs like the spleen or liver [51]. In the cancer vaccine field, the LNP adjuvant activity is partially responsible for the successful antitumor effect due to its capability to stimulate germinal center formation and T follicular helper (Tfh) cell response. In 2023, Zhu et al. showed that the optimization of the lipid ratio and the type of phospholipids used in LNP formulation enhanced the mRNA cargo delivery to DCs and their immune response activation against tumor cells. They found that nanoparticles with a zwitterionic helper lipid DOPE showed strong Th1-plus-Th2 responses, crucial for a coordinated antitumor effect. These findings emphasized the relevance of optimizing LNP formulation for success antigen-specific immune response [66]. Through a sialic acid (SA) modification, Tang et al. developed an mRNA vaccine that allows for efficient scape from early endosomes, preventing lysosomal entry and enabling the mRNA cargo to be simultaneously translated in ribosomes from the cytoplasm and endoplasmic reticulum [67]. SA targets the Siglec-1 receptor on DCs, so this modification enhances DC targeting. This study also proved that larger-sized nanoparticles prevent rapid clearance by the liver, allowing for sustained mRNA translation in DCs. ASSET (anchored secondary scFv [single-chain variable fragment] enabling targeting) is a cell-targeting platform based on the use of monoclonal antibody-coated LNP–siRNA or mRNA complexes. This approach uses a lipoprotein anchored to the LNP membrane that specifically interacts with the FC domain of an antibody [49,51,62,65]. In summary, the development of next-generation LNPs and other types of delivery carriers will further embrace mRNA-based therapies for a broad range of diseases.

### 3.4. Beyond Lipid Nanoparticles

Since 1980, viral vectors have been extensively employed in vaccine development due to their intrinsic ability to enter cells and deliver genetic material. However, several challenges still persist for this technology, including scalability issues in manufacturing, limited cargo capacity, and concerns regarding mutagenesis, toxicity, immunogenicity, and production costs [68]. Therefore, other alternatives to the use of viral vectors and LNPs have been explored as mRNA vaccine vehicles, which are described in the following.

#### 3.4.1. Protamine

The German company CureVac (Tübingen, Regierungsbezirk Tübingen, Germany) developed RNActive^®^ Technology, a two-component mRNA-based tumor vaccine. In 2011, they published the first study in mice that describes the combination of an antigenic and adjuvant in one vaccine. This vaccine consists of a mixture of a non-complexed sequence-optimized mRNA, which mediates antigen expression, and protamine complexed mRNA, the cationic compound that mainly acts as an adjuvant. Protamine is a small arginine-rich nuclear protein, which is known to stabilize DNA during spermatogenesis [69]. To date, there are seven studies registered in the clinical trials database (clinicaltrials.gov) from CureVac with RNActive^®^ Technology with a terminated or completed status (phase I/II) in non-small cell lung cancer (identifiers: NCT01915524 and NCT00923312), hormonal refractory prostate cancer (identifiers: NCT00906243 and NCT00831467), prostate cancer (identifiers: NCT02140138 and NCT01817738), and rabies (identifier: NCT02241135) patients.

#### 3.4.2. Arginine-Rich Peptide-Based mRNA Nanocomplexes

In 2014, the McCarthy group proposed a novel delivery system called RALA based on cell-penetrating peptides (CCPs), as they combine lower charge densities with strong membrane disruptive abilities [70]. The RALA peptide is a 30-mer cationic amphipathic peptide (with the sequence WEARLARALARALARHLARALARALRACEA) that enhances the delivery and transfection efficiency of mRNA. It is characterized by repeat units of arginine (R), alanine (A), leucine (L), and alanine (A). Arginine naturally occurs within DNA-binding/condensing motifs and serves as an effective transfection agent. Due to their increased internalization capabilities, arginine-rich peptide sequences have been broadly utilized to facilitate the transport of various gene delivery systems, including peptide dendrimers, liposomes, and polymer-based platforms. Studies have demonstrated that RALA mRNA nanocomplexes—which incorporate mRNA modified with the nucleosides pseudouridine and 5-methylcytidine—induce robust cytolytic T cell responses directed against the target antigen. These findings make RALA and other CCPs promising vehicles for mRNA vaccine delivery that are worth further investigation [70,71,72].

#### 3.4.3. Polymer-Based Nanoparticles for mRNA Delivery

Polymeric nanoparticles and dendrimers are two drug delivery systems. Polymeric nanoparticles are within a size range of 50–100 nm and synthesized from biodegradable polymers such as PLGA (poly lactic-co-glycolic acid) or PEG (polyethylene glycol). Although PLGA is approved by the FDA for the delivery of small-molecule drugs, it has not been approved for nucleic acids because, at neutral pH, it lacks the requisite positive charge to interact with the anionic RNA phosphodiester backbone. To overcome this limitation, researchers have conjugated cationic moieties to facilitate siRNA delivery in mice [49]. Also, the attachment of PEG chains to NPs, PEGylation, is used to diminish NP aggregation, improve stability, decrease clearance rates, and prolong systemic circulation in vivo. Despite nanoparticle improvement and PEGylation being used in numerous in vivo applications, there are reports of adverse effects, such as the production of antibodies against PEG and complement activation [73].

Dendrimers are highly branched macromolecules characterized by a central core, multiple layers, and terminal functional groups that can be conjugated with targeting ligands, among others. They exhibit a monodisperse, well-defined architecture and are typically between 1 and 10 nm in size. Dendrimers coupled to cationic groups—such as poly(amido-amine) (PAMAM) or poly-L-lysine (PLL)—can complex with RNA and facilitate its intracellular delivery. Their structure is often modified to shield RNA from enzymatic degradation and to promote endosomal escape [49]. Dendrimers have been successfully used to deliver RNA to the central nervous system [74], serve as intramuscular vaccines against pathogens like Ebola and H1N1 [75], and transport siRNA to hepatic endothelial cells [76].

In summary, LNPs, polymeric nanoparticles, dendrimers, and the use of other vehicles such as protamine or arginine-rich peptides are valuable tools in biomedicine, with their selection depending on the specific therapeutic requirement. Figure 2 schematizes the mRNA chemical modifications and the delivery systems discussed above.

## 4. New LNP Formulations: A Crucial Mix of Lipids to Fight Cancer

Targeting the correct cells to deliver a specific drug is crucial. Multiple lipid formulations have been developed to enhance the efficacy of nucleic acid delivery, improving cytotoxicity against tumor cells, enhancing DNA and mRNA stability, and reducing side effects. Marques et al. classified lipid-based nanoparticles into liposomes, solid lipid nanoparticles (SLNs), nanostructured lipid carriers (NLCs), and hybrid lipid–polymeric nanoparticles [77]. Below are some examples of innovations that have been made in the design of delivery systems based on LNPs in cancer therapies. A significant example of using LNPs, for example, is in the fight against brain cancers where small-molecule drugs struggle to cross the blood–brain barrier, leading to inadequate drug concentrations at the target site and reduced therapeutic effectiveness. Kuo and Liang [78] developed cationic solid lipid nanoparticles (CASLNs) with the inclusion of a monoclonal antibody (mAb) directed against the epidermal growth factor receptor (EGFR) carrying carmustatine (BCNU), a lipophilic chemotherapy drug used for brain tumors, Hodgkin’s and non-Hodgkin’s lymphoma, and multiple myeloma. The anti-EGFR/BCNU–CASLNs were cultivated with the human glioblastoma cell line U87MG, demonstrating effective delivery and antiproliferative efficacy against tumor growth.

Another monoclonal antibody used is the tumor necrosis factor (TNF)-related apoptosis-inducing ligand (TRAIL), also called Apo2 ligand (APO2L), a cytotoxic protein that activates rapid apoptosis in tumors but not in normal cells. De Miguel et al. reported an LNP coated with bioactive Apo2L/TRAIL named LUV-TRAIL, which improved TRAIL cytotoxic ability in non-small cell lung cancer (NSCLC) cell lines and in primary human tumor cells from patients with NSCLC. It also improved the antitumor effect of flavopiridol when combined with LUV-TRAIL [79]. More recently, Rahdari et al. [80] designed a novel solid lipid nanoparticle conjugated with a C-peptide (C-peptide–SLN) to deliver paclitaxel (PTX) in triple-negative breast cancer (TNBC). The C-peptide was derived from human endostatin, a protein capable of interacting with signaling molecules such as the integrin αvβ3 receptors overexpressed in TNBC cells. Therefore, the conjugation of LNPs with peptides that are ligands of these integrins could minimize off-target effects and decrease tumor cell adhesion, migration, and invasion. In this study, they reported that their conjugated nanoparticle (C-peptide–SLN–PTX) improved cytotoxic capacity, with an IC_50_ of 1.2 µg/mL compared to 3.4 µg/mL using the nanoparticle without C-peptide conjugation (SLN–PTX) and 8.9 µg/mL for free PTX against 4T1 carcinoma cells.

In another study on breast cancer, Yele et al. proposed the incorporation of programmed death ligand 1 (PD-L1)-binding peptides conjugated with PEGylated lipids into a lipid nanoparticle. The PD-L1 ligand is found in antigen-presenting cells (APCs) and tumor cells. The LNP encapsulates mRNA encoding the phosphatase and tensin homolog (PTEN) in an attempt to restore this tumor suppressor gene in PTEN-deficient triple-negative breast cancers (TNBCs). In vivo results showed autophagic cell death in 4T1 tumors, dendritic cell maturation, and T cell migration to the tumor site. Furthermore, the use of this conjugated nanoparticle exhibited an antimetastatic effect in advanced-stage PTEN-deficient TNBC models, suggesting its therapeutic potential [81].

The above are just a few examples of the many possibilities of using nanoparticle-based delivery systems. Moreover, nanotechnology-based drug delivery systems are being studied for surface conjugation with functional groups or biomolecules in a process known as functionalization. These conjugated ligands include antibodies, aptamers, peptides, polysaccharides and small molecules (e.g., folate) [77]. These options depend on the desired therapy, the type of cancer, and the target molecules to be reached.

## 5. Patents Registered in the Area of Therapeutic mRNA

Until 2022, the Chemical Abstracts Service (CAS) of the American Chemical Society (ACS) reported that a total of 2089 patents published worldwide, where 1129 patents are related to mRNA therapeutics for disease treatment, 977 are related to development of delivery technology, and 659 are related to vaccines. It is worth mentioning that the number of patent applications in these three types of technology accounts for 93.3% of the total patents, whereas the number of patents about mRNA modification technology is relatively small. CAS recognized 15 organizations with a high number of patents in mRNA therapeutics and vaccines, where companies are the main source of patents (11 out of 15). The U.S. and Germany are the top two countries with 550 and 214 patents, respectively, which account for about 67.7% of the total global patents in this area, and with 8 out of 15 and 4 out of 15 organizations, respectively. The top companies for mRNA vaccine research include Moderna (Cambridge, MA, USA), BioNTech (Mainz, Rhineland-Palatinate, Germany), Pfizer (New York City, NY, USA), and CureVac (Tübingen, Regierungsbezirk Tübingen, Germany) [82].

## 6. mRNA as Therapeutic Molecule in Vaccines

### Infectious Diseases Vaccines

mRNA vaccines are being developed for two major disease classes: infectious diseases and cancer. The search for the term “mRNA vaccines” in the clinical trial database https://clinicaltrials.gov/ (accessed on 2 January 2025) hits 582 studies with different statuses, and the majority (250) are classified as “completed”. Nowadays, the most advanced applications for mRNA therapeutics are vaccines for infectious diseases [50] (summarized in Table 2). Indeed, the first study involving an mRNA vaccine demonstrated an anti-influenza CD8+ T cell response in mice [2]. In the following, we discuss the use of mRNA in vaccines for infectious diseases, which have been developed to fight SARS-CoV-2, human cytomegalovirus, herpes simplex virus, among others.

## 7. Cancer Vaccines

Cancer is the second leading cause of death around the world, causing nearly ten million deaths and almost twenty million new cases in 2022 [93]. The search for a cure has been going on for more than 100 years, when William Halsted performed, in 1889, the first radical mastectomy to treat breast cancer [94]. Radiotherapy was used for the first time in 1896 to treat basal cell cancer [95], and chemotherapy was used in 1942 to treat lymphosarcoma [96]. Ever since, surgery, radio, and chemotherapy have become standard treatments for cancer. However, as a result of several key discoveries, new therapeutic options have been developed that take advantage of the immune system, such as the immune surveillance hypothesis proposed by Paul Ehrlich in 1909, based on the fact that the immune system acts like a patrol eliminating cells overexpressing specific antigens, thereby preventing tumor formation [97,98,99,100]. In the late 19th century, William Bradley Coley noted the tumor relapsing after fever and infections in 47 reports. In 1891, he started to inject living *Streptococcus pyogenes* in cancer patients looking for tumor regression through the feverish state. His experiments laid the foundations for today’s cancer immunotherapy [101,102]. Recently, the development of cancer vaccines has emerged as a promising field to fight cancer.

Cancer vaccines use tumor-associated antigens (TAAs) or tumor-specific antigens (TSAs), also termed neoantigens (NeoAgs) to stimulate the patient’s immune response [9,103]. Therapeutic cancer vaccines are different. Rather than teaching the immune system to recognize pathogens in advance of an infection, these vaccines use proteins produced by cancer cells to provoke a powerful immune response to existing tumors [104]. The principal aim of cancer vaccines is to provoke humoral and specific cellular immune responses to eradicate cancer cells and to avoid the tumor’s growth and regrowth. Cancer vaccines are classified according to whether they avoid some infections that lead to cancer development, named prophylactic or preventive cancer vaccines, such as the vaccine against human papillomavirus (HPV), which prevents cervical, anal, vaginal, vulva, penile, and oropharyngeal cancers; the vaccine for hepatitis B virus (HBV) for liver cancer [105,106]; or the recently IND (investigational new drug)-approved mRNA vaccine for Epstein–Barr virus-associated cancers [107]. The second type of vaccine is known as therapeutic when it is used in patients with cancer to stimulate the immune system against antigens in tumor cells. This type of vaccine can be divided into autologous and allogeneic, depending on whether the antigens came from the patient’s tumor; that is, they are personalized or from another patient’s tumor. Another type of classification is according to the platform used to deliver the antigens to the immune system, such as whole-cell, DCs, protein or polypeptide, and nucleic acid vaccines, which may be DNA or RNA vaccines.

Currently, two therapeutic cancer vaccines have been approved by the FDA to treat cancer: the *Bacillus* Calmette–Guerin vaccine, which is recommended for people with high risk of getting tuberculosis but shows effectivity treating early-stage bladder cancer [108], and Sipuleucel-T (Provenge), an autologous DC-based vaccine for advanced prostate cancer [109].

## 8. mRNA Cancer Vaccines

Due to the advances in mRNA technology, mRNA vaccines have emerged as a promising therapeutic strategy against cancer. Currently, mRNA vaccines are at the forefront of oncological research, with numerous clinical trials underway. The first mRNA vaccine clinical study published was a vaccine against melanoma [110]. Nowadays, the tumors targeted include pancreatic cancer, glioblastoma, melanoma, and lung cancer. They aim to stimulate the immune system to recognize self-proteins overexpressed in cancer cells, or self-proteins that have changed their expression pattern, like differentiation antigens or cancer germline/cancer testis antigens, or those proteins that have mutated upon malignant transformation, triggering the response that leads to cancer cell elimination. Additionally, vaccines could induce immunological memory against cancer antigens for immune protection against future infections or neoplastic cells [111,112,113,114,115]. Table 3 and Table 4 show the ongoing clinical trials of mRNA vaccines encoding different TAAs or neoantigens for different types of cancer. The first mRNA vaccine against cancer that initiated phase III was announced in July 2023, developed by the pharmaceutical companies Moderna and Merk together (identifier: NCT05933577). mRNA-4157 (also known as V940) is an experimental personalized cancer vaccine that encodes from 9 up to 34 different patient-specific neoantigens. Before the 1990s, the clinical trials might have used in vitro-transcribed mRNAs without any chemical modifications (e.g., pseudouridine or m1-pseudouridine), which might have induced immunogenic and inflammatory reactions. In the phase II KEYNOTE-942 trial 91% of patients received mRNA encoding the full 34 epitopes and showed a 44% decrease in the risk of post-surgical recurrence or death when injected with mRNA-4157 in combination with the anti-PD-1 drug pembrolizumab (Keytruda) as an adjuvant, in comparison with Keytruda alone. A phase III trial is under way, with final results expected in 2029 [116].

## 9. Classification

mRNA cancer vaccines could be classified as non-replicating mRNA (NRM), self-amplifying RNA (SAM), and transamplifying mRNA vaccines. Currently, SAM vaccines are in the preclinical stage, so the focus is mainly on non-replicating mRNA [9]. An NRM vaccine contains a conventional mRNA structure, including the cap, 5’- and 3’-UTRs flanking the ORF and the poly(A) tail. SAM vaccines are produced by gene engineering of alphaviruses, positive-stranded RNA viruses. They possess two ORFs: one encodes the target protein and the other encodes the viral replication component that directs the amplification of intracellular mRNA after delivery to the targeted cells, providing enhanced antigen expression and prolonged effectiveness even at lower doses [9,82,146,147,148]. Beissert et al. designed a transamplifying mRNA vaccine consisting of a replicase that amplifies RNAs in trans, allowing for shorter mRNA [146] and potentially simplifying large-scale production and manufacturing. However, it also introduces additional complexity in the delivery and production of two RNA-based therapeutics. Currently, clinical trials only include non-replicating mRNA vaccines [7].

### 9.1. Tumor Microenvironment (TME)

Tumor cells are never alone. They need an environment that shields them from the milieu. The tumor microenvironment is the complex entity that protects tumors from immune surveillance and therapeutic drugs, promotes their progression, and enhances their ability to invade other tissues [149]. The TME composition depends on the type of tumor and the stage of development but, broadly speaking comprises immune cells, such as APCs (like neutrophils, macrophages, and DCs), natural killer (NK) cells [150], tumor-associated macrophages (TAMs), and T and B cells; cancer cells; stromal cells, such as tumor-related fibroblasts and endothelial cells; the extracellular matrix (ECM); and soluble factors like cytokines, chemokines, and growth factors, which allow for communication between TME cells. There is a dichotomy in the cells that infiltrate the tumor, eliciting pro- and antitumorigenic effects [151]. For example, in the early stages of tumor establishment, TAMs can exhibit an M1-like polarization (tumor-suppressing effect) and, as the tumor progress, TAMs polarize to M2-like (tumor-promoting effect) [152]; TANs are also classified as tumor-suppressing (N1, with cytotoxic activity) or promoting (N2, which induce immunosuppression) phenotypes [153]. The TME surpasses the hypoxic and acidic microenvironment by promoting angiogenesis, which also promotes tumor progression by cellular proliferation induction and the epithelial–mesenchymal transition (EMT) [150]. Also, TME cells secrete soluble factors that disturb the basement membrane and remodel the ECM, increasing the ability of invasion and metastasis [154].

Understanding the TME composition and function is essential for developing effective therapeutic strategies. In the mRNA cancer vaccine field, Li and colleagues [155] investigated in mice a novel strategy to suppress tumor growth by intratumoral injections of a BNT162b2 COVID-19 mRNA vaccine, making tumoral cells express spike protein, awakening pre-existing memory immunity against this protein and killing those tumoral cell that express it. This approach lets the TME be reshaped by attracting immune cells. The combination of BNT162b2 with anti-PD-L1 therapy caused a better therapeutic impact, even in tumors typically less responsive to treatment.

### 9.2. Immune Evasion

Immunosurveillance is the process by which the immune system constantly looks for pre- or cancerous cells to destroy them. Nevertheless, cancer cells develop mechanisms against immunosurveillance, avoiding being “spotted” and killed by immune cells. The diminishment of antigen presentation is one of the immune evasion mechanisms where cancer cells reduce the expression of major histocompatibility complex (MHC) class I molecules, impairing antigen presentation and, consequently, avoiding recognition by CTLs [156]. Additionally, tumor cells foster an immunosuppressive environment by secreting cytokines with immunosuppressive actions, such as transforming growth factor β (TGF-β) and interleukin-10 (IL-10). Both inhibit pro-inflammatory cytokine production. The IL-10 pathway not only diminishes antigen presentation but also impairs phagocytosis [157], while TGF-β impairs T cell proliferation, differentiation, and effector functions in addition to suppressing the expression of MHC class II molecules, avoiding DC-presenting antigens [158]. The upregulation of immune checkpoint molecules occurs when cancer cells express proteins like cytotoxic T lymphocyte-associated protein 4 (CTLA-4) and programmed cell death ligand 1 (PD-L1) to bind their inhibitory receptors in immune cells, avoiding cancer cell destruction by T cells [159]. Another mechanism for immune escape is the recruitment of immunosuppressive cells like myeloid-derived suppressive cells, associated with inhibiting T cell activation and proliferation, as well as promoting regulatory T cell (Tregs) expansion; M2-like TAMs; N2 TANs; Tregs, which inhibit CTL and NK cell activity by secreting TGF-β and IL-10, maintaining the immunosuppressive TME; and tumor-associated DCs (tDCs), which usually have a tolerogenic phenotype, promote Treg expansion, and inhibit CTL activation [160].

The TME and immune evasion are closely interconnected and can be targeted using mRNA vaccines, either alone or in combination with other immunotherapies. Current research explores approaches such as immune checkpoint inhibitors and monoclonal antibodies. Additionally, mRNA vaccines have the potential to remodel the TME by regulating the balance between M1 and M2 macrophages, modulating cytokine secretion by various T cell subpopulations [161], or encoding immunostimulatory molecules [162], thereby overcoming the immunosuppressive state.

## 10. mRNA Cancer Vaccine Mechanisms

mRNA cancer vaccines stimulate the immune system by expressing a protein overexpressed in cancer cells. Then, the immune system will eliminate those cells expressing that protein (Figure 3). Therapeutic mRNA can be encapsulated in LNPs to avoid degradation and facilitate cellular uptake into APCs by endocytosis in the vaccine site [163]. There, the mRNA will escape from endosomes due to the acidic environment [164] and will be translated by ribosomes into the therapeutic protein. Then, the target protein is processed into antigen–MHC I/II molecules to be expressed on the cell surface. When mRNA is translated into target protein in cytoplasm, proteins will be ubiquitinated and degraded by proteasome into peptides, which are translocated to the endoplasmic reticulum (ER) where they bind to MHC class I molecules. Subsequently, peptide–MHC I complexes are transported to the cell surface where they can present antigenic peptides to CD8+ cytotoxic lymphocytes [165]. On the other hand, regarding APCs’ endocytic extracellular cancer antigens, these are degraded in peptide fragments by endosomal/lysosomal compartments, where they are loaded into MHC class II molecules synthetized in the ER. Peptide–MHC II complexes are transported to the cell surface, where they will be presented to CD4+ helper T cells that will secrete cytokines, such as IL-12 and IL-18, to enhance the activation and proliferation of CD8+ T cells and B cells that produce specific-antigen antibodies, thereby amplifying the immune response against tumor cells [166]. In this moment, the translated protein stimulates DCs by toll-like receptors (TLRs) [27,167]. Once activated, DCs are transported to lymph nodes where they activate T cells by the antigen–MHC I/II complexes binding to the T cell receptor (TCR) on the surface of CD8+/ CD4+ T cells, which leads to the binding of costimulatory signaling molecules to receptors and cytokines, like interferon-γ (IFN-γ), and interleukin 12 (IL-12) binding to cytokine receptors on the T cells [108,109]. Additionally, IL-12 secreted by CD4+ T cells promotes CD8+ T cell amplification. Subsequently, activated T cells move toward the tumor, where they infiltrate the tissue by chemokine action to amplify the antitumor response of their secreted effectors [103,168,169]. Additionally, mRNA vaccines can provoke the immune response through TLR7 and 8 present in endosomes, due to their ability to recognize single-stranded RNA (ssRNA) [170,171].

## 11. Neoantigens: Mutations Turned into a Vaccine

Personalized neoantigen-based vaccines use so-called neoantigens, also known as tumor-specific antigens (TSAs), which arise from somatic mutations bearing single-nucleotide variants (SNVs) or are derived from DNA insertions, deletions, or genomic rearrangements, resulting in an alteration in gene expression [172]. The mutated protein (or neoantigen) can be injected in protein form or as gene therapies, which results in the expression of tumor antigens by antigen-presenting cells (APCs). The antigens are presented to naïve T cells, increasing the likelihood that cytotoxic T cells specific to tumor neoantigens will become activated and kill cancer cells. However, some challenges are encountered by neoantigen vaccines. These types of vaccines are not meant to be reactive to exhausted T cells, so it has been proposed to combine them with already existing treatments including immune checkpoint inhibitors, chemotherapy, and radiation therapy [173]. The high level of personalization increases both the cost of the therapy and the time required to administer the treatment. Since not all mutated proteins are immunogenic, a large number of neoantigens must be included in the putative vaccine. For example, Moderna’s melanoma vaccine (mRNA 4157) includes at least 34 different neoantigens obtained from patient tumor samples. Figure 4 depicts the selection of neoantigens for personalized vaccine development.

Finally, the continuous change in antigen expression may be the greatest challenge in these vaccines, which highlights the importance of studying the tumor environment and immune escape mechanisms to develop, in the near future, more potent and effective personalized cancer vaccines.

## 12. Outlook

mRNA technology might revolutionize immunotherapy against cancer because mRNA cancer vaccines offer a highly promising approach for targeting TAAs or TSAs, thereby stimulating the immune system to destroy malignant cells. However, before the revolution moves forward, several challenges must be addressed to achieve clinical success. These include the identification of optimal tumor antigens; the design of the right structure of mRNA to properly express the cancer antigens; the improvement of mRNA nanocarrier formulations and size for efficient delivery to tumor cells; the stimulation of a sufficient immune response without inducing harmful inflammation; ensuring precise organ-specific delivery; and surpassing tumor immunosuppressive and resistance mechanisms such as the secretion of immunosuppressive cytokines or the activation of regulatory cell subpopulations. Finally, combined treatments of chemo- or radiotherapies along with mRNA vaccines and/or checkpoint inhibitors should be tested in the future. Sooner or later, these challenges will probably be solved.

## Figures and Tables

**Figure 1 genes-16-00376-f001:**
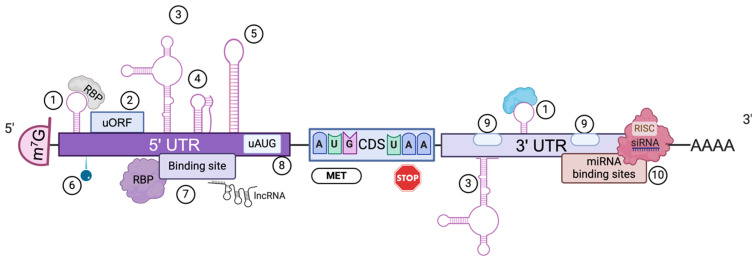
Regulatory elements of eukaryotic mRNA. The structure and regulatory elements of mRNAs are essential for its function in gene expression. Eukaryotic mRNAs have a cap structure at the 5’ end followed by a 5’-untranslated region (5’-UTR), where we may find different regulatory elements: 1. Ribonucleoprotein (RNP) complex. 2. Upstream open reading frames (uORFs). 3. Hairpin loops. 4. Pseudoknots. *5.* Internal ribosome entry sites (IRESs). 6. RNA modifications. 7. Binding sites. 8. Upstream AUG codons. The 3’-UTR may possess the following: 9. Poly(A) signals (PASs). 10. microRNA-binding sites. *RBP*, RNA-binding protein; *AUG*, translation initiation codon; MET, methionine; CDS, coding sequence; UAA; translation stop codon; siRNA, small interference RNA; AAAA, poly(A) tail.

**Figure 2 genes-16-00376-f002:**
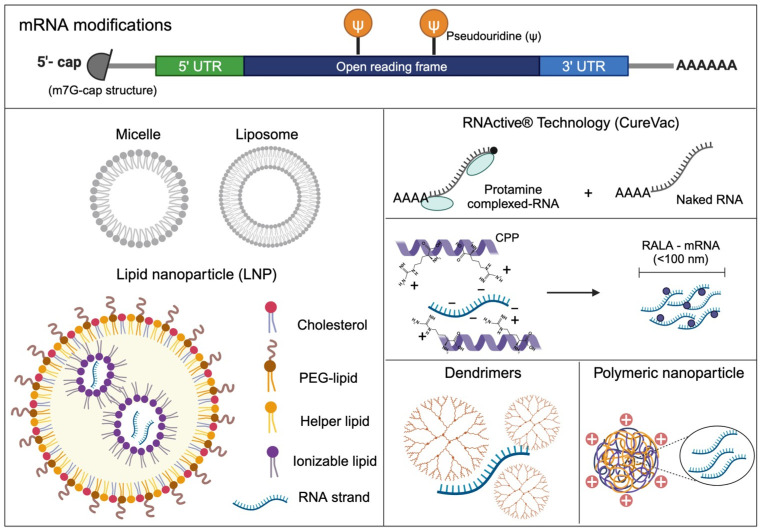
mRNA structure and modifications to ensure its stability, translation, and efficiency along with some delivery systems mentioned in this review. PEG, polyethylene glycol; CPP, cell-penetrating peptide; RALA, repeat units of arginine (R), alanine (A), leucine (L), and alanine (A).

**Figure 3 genes-16-00376-f003:**
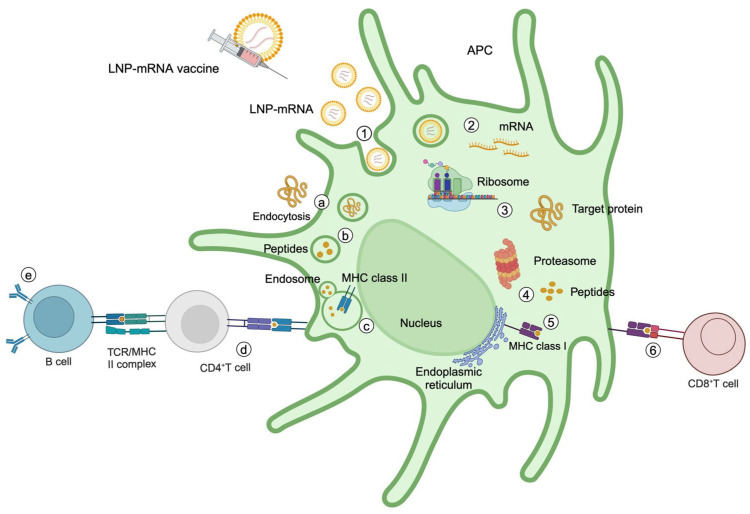
Mechanisms of action of mRNA cancer vaccines. (1) Endocytosis internalizes therapeutic mRNA within lipid nanoparticles (LNPs) into an antigen-presenting cell (APC). (2) Therapeutic mRNA escapes from the endosome and (3) is translated into protein by the ribosome. (4) The target protein is degraded by the proteasome complex into antigenic peptides. (5) The antigenic epitopes are loaded onto MHC class I molecules in the endoplasmic reticulum and move toward the cell surface. (6) MHC class I molecules present antigenic peptides to CD8+ T cells. An alternative process occurs when the protein is secreted and internalized into an APC by endocytosis (a) and is degraded into antigenic peptides in endosomes (b), where they are loaded onto MHC class II molecules (c). Antigenic peptides are presented to CD4^+^ T cells, which activates them (d). CD4^+^ T cells stimulate B cells to produce neutralizing antibodies against the target protein present in cancer cells (e).

**Figure 4 genes-16-00376-f004:**
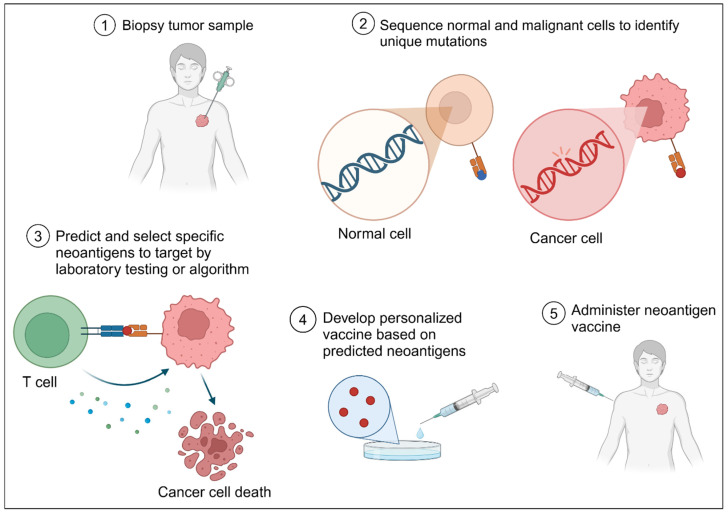
Neoantigen vaccine production. A simplified scheme of the whole process is depicted.

**Table 1 genes-16-00376-t001:** Typical composition and function of lipids in LNPs for mRNA delivery ^a^.

Lipid		Function ^a^	Proportion (%)	Examples	References
Ionizable cationic lipids (ICLs)		Enables proper mRNA encapsulation and facilitates mRNA escape into cytosol. Key role in intracellular delivery.	30–50	MC3 (DLin–MC3–DMA); KC2 (Dlin–KC2–DMA); Lipid H (SM-102); ALC-0315	[51,63,64]
Structural lipids	Phospholipid (or helper lipids)	Lipid bilayer formation and core–shell structure, keeping LNP integrity. Facilitates mRNA intracellular delivery by increasing LNP fusion with cellular and endosomal membranes.	20–50	DSPC; DOPE	[51,65]
	Sterol	Enhance particle membrane stability and modulate membrane fluidity for fusion and cellular uptake.	10–20	Cholesterol	[63]
PEG–lipids		Stabilize nanoparticles, preventing aggregation; prolong circulation time due to opsonization reduction by PEG corona, which also allows for mononuclear phagocytic system evasion.	~1.5	PEG–DMG, PEG–DSPE	[51,63]

^a^ Modified from Liu et al. 2024 [62].

**Table 2 genes-16-00376-t002:** Relevant mRNA vaccines in infectious diseases.

International Publication Number	Publication Date	Applicant	Title	Summary	References
WO 2013/055905	April 2013	Novartis (Basel, Switzerland)	Recombinant self-replicating polycistronic RNA molecules	The recombinant polycistronic self-replicating RNA molecules co-deliver 4 or more proteins to cells, triggering widespread and strong immune responses convenient for vaccine development.	[83]
WO 2021/159040	August 2021	Moderna TX, Inc. (Cambridge, MA, USA)	SARS-CoV-2 mRNA domain vaccines	m1Ψ-modified mRNA-LNP vaccine comprising a fusion protein of spike protein domains from SARS-CoV-2 and an influenza hemagglutinin transmembrane domain.	[84]
WO 2021/159130	August 2021	Moderna TX, Inc. (Cambridge, MA, USA)	Coronavirus RNA vaccines and methods of use	A mRNA-LNP vaccine comprising an ORF encoding SARS-CoV-2 spike protein to induce a neutralizing antibody response against this protein.	[85]
WO 2021/255270	December 2021	Ziphius vaccines (Zwijnaarde, Belgium); Universiteit Gent. Belgium	Self-amplifying SARS-CoV-2 RNA vaccines	An alphavirus-self-amplifying mRNA vaccine in combination with sequences of the spike and nucleocapsid proteins from SARS-CoV-2 able of induce a robust protection against SARS-CoV-2 variants.	[86]
WO 2017/070613	April 2017	Moderna TX, Inc. (Cambridge, MA, USA)	Human cytomegalovirus vaccine	HCMV (human cytomegalovirusay ) RNA vaccines of one or more ORFs encoding antigens from this virus.	[87]
WO 2021/204179	October 2021	Suzhou Abogen Biosciences Co., LTD. (Suzhou, China)	Nucleic acid vaccines for coronavirus	Therapeutic DNA or RNA with one or more modified nucleotides ψ, m1ψ, and 5mC for the management, prevention, and/or treatment of infectious diseases caused by coronavirus.	[88]
WO 2021/226436	November 2021	Translate Bio, Inc. (Boston, MA, USA); Sanofi Pasteur Inc. (Paris, France)	Optimized nucleotide sequences encoding SARS-CoV-2 antigens	Optimized mRNA sequence encoding SARS-CoV-2 antigens encapsulated by LNP. Suitable for vaccine use for therapy or prophylaxis of infections from by β-coronaviruses.	[89]
WO 2017/070623	April 2017	Moderna TX, Inc. (Cambridge, MA, USA)	Herpes simplex virus vaccine	Herpes simplex virus (HSV) mRNA-LNP vaccines comprising at least one short sequence from HSV antigens fused to a signal peptide. These contain at least one chemical modifications.	[90]
WO 2021/160346	August 2021	Institute Pasteur. (Paris, France).	Nucleic acid vaccines against the SARS-CoV-2 coronavirus	Optimized nucleic acid vaccines encoding SARS-CoV-2 spike protein with one or more mutations and a Kozak sequence.	[91]
WO 2022/171182	August 2022	Stermirna Therapeutics Co., LTD (Shanghai, China)	Vaccine reagent for treating or preventing coronavirus mutant strain.	A vaccine comprising the S1 and the S2 subunits from SARS-CoV-2 spike protein for the prevention or treatment of coronavirus infection.	[92]

**Table 3 genes-16-00376-t003:** mRNA cancer vaccines in active, completed, or terminated clinical studies ^a^.

Vaccine Name/ Sponsor	NCT Number (Phase)	Cancer Type	mRNA Encoding Antigen	Status	Clinical Trial Outcome	References
Autogene cevumeran (also known as BNT122, RO7198457)/ Genentech, Inc. (San Francisco, CA, USA).	NCT05968326 (II)	PDAC	Patient-specific cancer (neoantigens)	Recruiting	In a phase I trial, it was well tolerated and prompted the development of strong, de novo neoantigen-specific T cells in 50% of patients. Responder patients had a longer median recurrence-free survival compared to non-responder patients.	[117]
mRNA 4157/ ModernaTX, Inc. (Cambridge, MA, USA).	NCT03897881 (II)	Melanoma	Individualized neoantigen therapy	Recruiting	Phase 2b results showed that mRNA-4157 vaccine combined with pembrolizumab extended recurrence-free survival vs. pembrolizumab alone (18 months, 79% vs. 62%) in 157 patients.	[115]
mRNA 4359/ ModernaTX, Inc (Cambridge, MA, USA).	NCT05533697 (I/II)	Melanoma; NSCLC	PD-L1 and IDO1	Recruiting	No results posted.	[118]
BI1361849 (CV9202)/ CureVac (Tübingen, Regierungsbezirk Tübingen, Germany)	NCT01915524 (I)	NSCLC	NY-ESO-1 ^a^, MAGE-C1, MAGE-C2, survivin, 5T4, and MUC-1 ^a^	Terminated; slow recruitment	Out of 25 patients, one showed a partial response when treated with pemetrexed maintenance, and 46.2% had stable disease as their best overall response.	[119]
CV9103/CureVac	NCT00831467 (I/II)	Hormonal refractory prostate cancer	PSA ^a^, PSCA ^a^, PSMA ^a^ and STEAP1 ^a^	Completed	Median overall survival of 29.3 months.	[120]
CV9104/CureVac	NCT01817738 (I/II)		PSA, PSCA, PSMA, STEAP1, Muc-1, and survivin	Terminated	Median overall survival of 35.5 months vs. 33.7 compared with placebo; did not show statistical significance difference.	[121,122]
SW 1115C3/Stemirna Therapeutics (Pudong New Area, Shanghai, China)	NCT05198752 (I)	Solid tumors	Neoantigen mRNA personalized cancer vaccine	Unknown status	No results posted.	[123]
AGS-003/Argos Therapeutics (Durham, NC, USA).	NCT00678119 (II)	Renal cell carcinoma	Autologous total tumor RNA and synthetic CD40L RNA	Completed	62% of patients (intermediate- and poor-risk mRCC) had partial clinical benefits. Enrollment was terminated early. 33% survived for at least 4.5 years, 24% survived for more than 5 years, and two patients who remained progression-free with durable responses for more than 5 years at the time of this report.	[124]
GRNVAC1/Asterias Biotherapeutics, Inc (Blacksburg, VA, USA)	NCT00510133 (II)	Acute Myelogenous Leukemia	hTRT ^a^	Completed	No results posted.	[125]
H-42434/Baylor College of Medicine	NCT04157127 (I)	Pancreatic adenocarcinoma	Tumor cell lysate and RNA	Active; not recruiting	No results posted.	[126]
Ag-mRNA- DC-999brain/Guangdong 999 Brain Hospital	NCT02709616 (I) NCT02808364 (I) NCT02808416 (I)	Glioblastoma/NSCLC ^a^	Personalized TAA panels containing different TAAs	Completed	Ten patients were enrolled (5 NSCLC and 5 GBM). The median survival time was 17 months for the lung cancer patients and 19 months for the GBM patients. Survivin was a commonly identified TAA.	[127]
UR1121/Inge Marie Svane, Herlev Hospital	NCT01446731 (II)	CRMPC ^a^	PSA ^a^, PAP ^a^, Surviving and Htert ^a^ mRNA	Completed	The specific-antigen responses were similar in patients receiving either docetaxel monotherapy or combination therapy. The toxicity from the vaccine was confined to local reactions.	[128]
DC-005/Oslo University Hospital	NCT01197625 (I/II)	Prostate cancer	mRNA from primary prostate cancer tissue, hTERT ^a^ and survivin	Active; not recruiting	55% of patients were BCR-free over a median of 96 months while 45% developed BCR during or after vaccination regimen. Patients who developed BCR maintained stable disease up to 99 months.	[129]
TriMix-DC- MEL/Bart Neyns, Universitair Ziekenhuis Brussel	NCT01302496 (II)	Malignant melanoma stage III and IV	MAGE-A3 ^a^, MAGE-C2, tyrosinase and gp100	Completed	Treatment elicits strong CD8+-cell response in 80% of late stages melanoma patients.	[130]
Lipo-MERIT or FixVac (BNT111) BioNTech SE	NCT02410733 (I)	Melanoma	NY-ESO-1 ^a^, tyrosinase, MAGE-A3, and TPTE ^a^	Completed	Either alone or in combination with an α-PD1 induces activation of robust CD4+ and CD8+ T cell immunity against the vaccine antigens.	[131]
IVAC_W_bre 1_uID and IVAC_M_uID or (TNBC-MERIT) BioNTech SE	NCT02316457 (I)	TNBC ^a^	Encoding neoepitopes derived from up to 20 cancer mutations determined by NGS	Completed	All patients exhibited specific CD4+ and/or CD8+ T cell responses to 1 to 10 of the vaccine neoepitopes detected by IFN-γ ELISpot, ex vivo or following in vitro stimulation.	[132]
BNT113/BioNTech SE	NCT04534205 (II)	Metastatic HNSCC ^a^	HPV-16 ^a^ oncoproteins E6 and E7	Recruiting	No results posted.	[133]
W_ova1 Vaccine/University Medical Center Groningen	NCT04163094 (I)	Ovarian cancer	3 ovarian cancer-related TAA	Terminated	No results posted.	[134]

^a^ Abbreviations: PDAC, pancreatic ductal adenocarcinoma; NSCLC, non-small cell lung carcinoma; PD-L1, program death ligand 1; IDO1, indoleamine 2,3-dioxygenase 1; NY-ESO-1, New York-ESO 1; MAGE, melanoma-associated antigen; 5T4, human oncofetal antigen 5T4; MUC-1, mucin-1; *PSA*, prostate specific antigen (*KLK3*, kallikrein-related peptide 3); PSCA, prostate stem cell antigen; PSMA, prostate-specific membrane antigen (*FOLH1*, folate hydrolase); STEAP1, six transmembrane epithelial antigen of the prostate; hTRT, human telomerase reverse transcriptase; CRMPC, castration resistant metastatic cancer prostate; PAP, prostatic acid phosphatase; PFS, progression-free survival; DSS, disease-specific survival; BCR, biochemical relapse; IPI, ipilimumab; OS, overall survival; TPTE, trans-membrane phosphatase with tensin homology; TNBC, triple negative breast cancer; NGS, next-generation sequence; HPV, human papilloma virus; HNSCC, head and neck squamous cell carcinoma.

**Table 4 genes-16-00376-t004:** The most relevant mRNA vaccines in cancer.

International Publication Number	Publication Date	Applicant	Title	Summary	References
WO 2021/155149	August 2021	Genentech (San Francisco, CA, USA); BioNTech SE (Mainz, Germany); F. Hoffmann-La Roche, (Basel, Switzerland)	Methods of inducing neoepitope-specific T cells with a PD-1 axis-binding antagonist and an RNA vaccine.	Provides methods to induce neoepitope-specific CD8+ T cells in an individual using an RNA vaccine with different doses (100, 75, 50, 38, and 25 μg) in combination with an anti PD-1. The RNA sequences encode one or more neopeptides resulting from cancer-specific somatic mutations present in patient tumor samples.	[135]
WO 2015/024664	February 2015	CureVac AG (Tübingen, Germany)	Composition and vaccine for treating prostate cancer	It refers to an mRNA vaccine that encodes a combination of antigens (PSA, PSMA, PSCA, STEAP, MUC1 and PAP) to elicit an immune (adaptive) response preferably in patients with prostate adenocarcinoma, locally limited, locally advanced, metastatic, castration-resistant (hormone refractory), metastatic castration-resistant, and non-metastatic castration-resistant prostate cancers.	[136]
WO 2012/019168	February 2012	Moderna Therapeutics, Inc. (Cambridge, MA, USA)	Engineered nucleic acids and methods of use thereof	The modified mRNAs (mmRNAs) encode melanocyte-stimulating hormone (MSH), insulin, and G-CSF, and compositions and methods for its delivery into cells to modulate protein expression are included.	[137]
WO 2020/097291	May 2020	Moderna Therapeutics, Inc. (Cambridge, MA, USA)	RNA cancer vaccines	The mRNA vaccine is formulated into a lipid nanoparticle and comprises one mRNA having one open reading frame encoding 3 to 50 or 20 to 40 or 30 to 35 or 34 peptide epitopes. Each of the peptide epitopes are portions of personalized cancer antigens or portions of cancer hotspot antigens. Different doses were investigated from 0.4 tp 5.0 mg. It includes methods for preparation such as lipid nanoparticle composition.	[138]
WO 2020/141212	January 2020	eTheRNA Immunotherapies NV (Niel, Belgium)	mRNA vaccine	It includes a combination of one or more mRNA molecules encoding at least one functional immunostimulatory protein: CD40L, CD70, and caTLR4; and an anti-PD-1 (optionally also in the form of an mRNA molecule).	[139]
WO 20220/08519	January 2022	BionTech (Mainz, Germany); TRON—Translationale Onkologie an der Universitätsmedizin der Johannes Gutenberg-Universität Mainz Gemeinnützige GMBH, (Mainz, Germany).	Therapeutic RNA for HPV-positive cancer	The mRNA vaccine was designed to treat HPV-positive cancers (anogenital, cervical and penile cancers and head and neck squamoues cell carcinoma [HNSCC]. Some results include a reduction in tumor size, prolonged time to progressive disease, and/or protection against metastasis resulting in an extension of survival time.	[140]
WO 2015/024666	February 2015	CureVac AG (Tübingen, Germany)	Composition and vaccine for treating lung cancer	The antigens included in this vaccine (also named CV9202) are 5T4 (Trophoblast glycoprotein, TPBG), survivin (Baculoviral IAP repeat-containing protein 5; BIRC5), NY-ESO-1 (New York esophageal squamous cell carcinoma 1, CTAG1 B), MAGE-C1 (melanoma antigen family C1), MAGE-C2 (melanoma antigen family C2), and MUC1 (mucin-1) to effectively stimulate an (adaptive) immune response to treat lung cancer.	[141]
WO2012159643	November 2012	BionTech (Mainz, Germany); TRON—Translationale Onkologie an der Universitätsmedizin der Johannes Gutenberg-Universität Mainz Gemeinnützige GMBH, (Mainz, Germany).	Individualized vaccines for cancer	The patent includes the invention of a personalized vaccine, in which neoepitope sequences are taken from patient tumor samples; this vaccine can be used as a naked vaccine in formulation buffer or encapsulated (e.g., nanoparticles, liposomes) for direct injection (e.g., lymph nodes, s.c., i.v., i.m.). Also, it can be used for in vitro transfection (e.g., dendritic cells) for adoptive transfer.	[142]
WO 2015/014869	February 2015	BionTech (Mainz, Germany).; TRON—Translationale Onkologie an der Universitätsmedizin der Johannes Gutenberg-Universität Mainz Gemeinnützige GMBH (Mainz, Germany).	Tumor antigens for determining cancer therapy	The vaccine contains the antigens CXorf61, CAGE1, and PRAM to treat breast cancer (particularly triple-negative breast cancer).	[143]
WO 2022/009052	January 2022	Janssen Biotech Inc. (Raritan, NJ, USA)	Prostate neoantigens and their uses	The self-replicating RNAs encode prostate neoantigens. This method aims to activate vaccine-specific CD8+ and CD4+ T cells, enhancing the body’s ability to combat cancer cells effectively through increased production of TNF-α and IFN-γ.	[144]
WO 2022/081764	May 2022	RNAimmune Inc. (Gaithersburg, MD, USA)	Pan-RAS mRNA cancer vaccine	The vaccine includes mRNAs that express cancer neoantigens, derived from mutated human RAS genes. They are formulated with pharmaceutically acceptable carriers such as 1,2-dioleoyl-3-trimethylammonium propane (DOTAP).	[145]

## Data Availability

No new data were created or analyzed in this study. Data sharing is not applicable to this article.

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
