# Peer review of "The Bright Future of mRNA as a Therapeutic Molecule"

_genes, 2025, doi:10.3390/genes16040376_

Round 1
Reviewer 1 Report
Comments and Suggestions for Authors
LNP is not my speciality so I cannot comment on that section. But the authors made a great in gathering previous and current mRNA clinical trials and the major patents filed on specific RNA therapeutics. I only have a few points that could help improve the quality of this review article:
1. The last and second last paragraph of section 3 (line 88-122) can use some polishing.
2. section 3.1. mRNA Modifications. "Modifications" in RNA and DNA in general refer to chemical modifications, especially methylation. I suggest the authors to use a different term here to describe the nucleotide sequence engineering in this section.
3. Line 130, the comment about stable secondary structures should be avoided in the ORF is not universally true. There are studies showing that some level of structures improve translation.
4. Line 131-132, the comment about the longer poly(A) tail the higher translation efficiency is also not true. There tends to be a limit after which more A's does not further improve translation.
5. Section 5, line 333-335: instead of bringing up the potential application in protein replacement therapy, I suggest the authors give some context for Table 3.
6. Section 6, line 396-410: The authors should find out which of the examples shown in Table 4 used modified nucleotides (eg. pseudouridine, m1-pseudouridine, etc) and which used natural nucleotides. I suspect a lot of the clinical trials done in the 90's used IVT RNA without chemical modifications. Those synthetic mRNA might have induced severe immunogenic reactions and led to termination of the clinical trials.
7. Section 8: line 431: Most if not all self-amplifying RNA vaccines currently in development are based on alphavirus. I suggest the authors point that out instead of referring simply to "positive-strand RNA viruses."
8. The article appears to have ended abruptly after discussing mRNA cancer vaccine mechanisms in Section 9. Is there supposed to have a concluding message?
9. Reference #2 is not available for review.
Comments on the Quality of English LanguageLine 97: "activate" instead of "active"
Lines 97-122 needs polishing.
Line 398: "However, none of them had managed to advance into phase III" is very vague and is not followed up by elaboration of the context of the statement.
Author Response
Comments and Suggestions for Authors
LNP is not my speciality so I cannot comment on that section. But the authors made a great in gathering previous and current mRNA clinical trials and the major patents filed on specific RNA therapeutics. I only have a few points that could help improve the quality of this review article:
- The last and second last paragraph of section 3 (line 88-122) can use some polishing.
Response: Done.
- section 3.1. mRNA Modifications. "Modifications" in RNA and DNA in general refer to chemical modifications, especially methylation. I suggest the authors to use a different term here to describe the nucleotide sequence engineering in this section.
Response: Agree. We have changed the term “modifications” to “nucleotide sequence engineering”. Page 5, lanes 21-22.
- Line 130, the comment about stable secondary structures should be avoided in the ORF is not universally true. There are studies showing that some level of structures improve translation.
Response: Agree. Page 5, lanes 28-30.
- Line 131-132, the comment about the longer poly(A) tail the higher translation efficiency is also not true. There tends to be a limit after which more A's does not further improve translation.
Response: Agree. Page 5 lane 33, and page 6 lanes 1-3.
- Section 5, line 333-335: instead of bringing up the potential application in protein replacement therapy, I suggest the authors give some context for Table 3.
Response: Done. Page 13, lanes 21-29. Table 3 is now Table 2.
- Section 6, line 396-410: The authors should find out which of the examples shown in Table 4 used modified nucleotides (eg. pseudouridine, m1-pseudouridine, etc) and which used natural nucleotides. I suspect a lot of the clinical trials done in the 90's used IVT RNA without chemical modifications. Those synthetic mRNA might have induced severe immunogenic reactions and led to termination of the clinical trials.
Response: Agree. However, we could not find this information. We have commented this point in page 17, lanes 27-29. Table 4 is now Table 3.
- Section 8: line 431: Most if not all self-amplifying RNA vaccines currently in development are based on alphavirus. I suggest the authors point that out instead of referring simply to "positive-strand RNA viruses."
Response: Done. Page 15.
- The article appears to have ended abruptly after discussing mRNA cancer vaccine mechanisms in Section 9. Is there supposed to have a concluding message?
Response: We have included an “Outlook” section at the end of the article. Page 30.
- Reference #2 is not available for review.
Response: It is now fixed.
Comments on the Quality of English Language
Line 97: "activate" instead of "active"
Lines 97-122 needs polishing.
Response: Done.
Line 398: "However, none of them had managed to advance into phase III" is very vague and is not followed up by elaboration of the context of the statement.
Response: Agree. Because we lack more information, we have deleted this sentence. Page 17, lane 22.
Reviewer 2 Report
Comments and Suggestions for Authors
The authors aim at reviewing basic concepts, key historical aspects, and recent research of the mRNA as a therapeutic molecule to fight infectious diseases and cancer and presenting a patent landscape. The issue is that some statements are not correct. For example the first sentence: “Back in 1990, Wolff and collaborators proposed the concept of messenger (m) RNA 22 vaccine [1].”. No Wolff et al did not propose here the concept of mRNA vaccine. They did not measure immune responses against the encoded antigens in mRNA-injected mice. It is Martinon 1993 (European Journal of Immunology) that shows vaccination with mRNA for the first time. The second sentence “However, it was until 2020 that the COVID-19 pandemic prompted the development of vaccines based on mRNA”. No, mRNA vaccines have been intensively developed for example by CureVac created in 2000, BioNTech created in 2008 and Moderna created in 2010. Third sentence “The technological innovations that made it possible paved the road for the future development of vaccines against other maladies, including cancer and infectious diseases.”. No, it is mostly the development of mRNA vaccines against cancer (CureVac and BioNTech) that has made possible the COVID vaccines, not the opposite. “They found that TLR7 and TLR8 sense single-stranded RNA (ssRNA)”: who is “they”? It was not made by Kariko and Weissman. Contrarily to what is said, mRNA vaccines containing non-modified nucleotides (ACGU, no (m1)pseudoU) are safe and work well: mRNA vaccine against COVID19 by CureVac (unmodified) was shown to be safe and efficient even if the company decided not to commercialise it, the Arcturus vaccine is non modified and approved (Japan), all anti-cancer vaccines of BioNTech are non-modified…. Line 372 “the recently approved mRNA vaccine for Epstein-Barr virus-associated cancers”, as far as I know there is no approved mRNA vaccine against EBV. Line 386” Recent advancements in mRNA technology and the COVID emergency have brought the eyes to mRNA vaccines in the last years, and mRNA vaccines have emerged as a promising therapeutic strategy against cancer.”, no mRNA vaccines against cancer have been tested and optimised before 2020 and thanks to this the COVID vaccines could be developed. The first mRNA vaccine clinical study published was an anticancer (melanoma) vaccine by Weide et al 2008. Line 398 “However, none of them had managed to advance into phase III.”, no there is one in phase III as mentioned next line!
Author Response
Reviewer #2
The authors aim at reviewing basic concepts, key historical aspects, and recent research of the mRNA as a therapeutic molecule to fight infectious diseases and cancer and presenting a patent landscape. The issue is that some statements are not correct. For example the first sentence: “Back in 1990, Wolff and collaborators proposed the concept of messenger (m) RNA 22 vaccine [1].” No Wolff et al did not propose here the concept of mRNA vaccine. They did not measure immune responses against the encoded antigens in mRNA-injected mice. It is Martinon 1993 (European Journal of Immunology) that shows vaccination with mRNA for the first time.
Response: Agree. Page 3, lane 2-4.
The second sentence “However, it was until 2020 that the COVID-19 pandemic prompted the development of vaccines based on mRNA”. No, mRNA vaccines have been intensively developed for example by CureVac created in 2000, BioNTech created in 2008 and Moderna created in 2010.
Response: Agree. Page 3, lanes 8-11.
Third sentence “The technological innovations that made it possible paved the road for the future development of vaccines against other maladies, including cancer and infectious diseases.”. No, it is mostly the development of mRNA vaccines against cancer (CureVac and BioNTech) that has made possible the COVID vaccines, not the opposite.
Response: Agree. Page 3, lanes 8-12.
“They found that TLR7 and TLR8 sense single-stranded RNA (ssRNA)”: who is “they”? It was not made by Kariko and Weissman. Contrarily to what is said, mRNA vaccines containing non-modified nucleotides (ACGU, no (m1)pseudoU) are safe and work well: mRNA vaccine against COVID19 by CureVac (unmodified) was shown to be safe and efficient even if the company decided not to commercialise it, the Arcturus vaccine is non modified and approved (Japan), all anti-cancer vaccines of BioNTech are non-modified….
Response: Agree. We modified the sentence accordingly. Page 6 lanes 21-27, and page 7 lanes 2-6.
Line 372 “the recently approved mRNA vaccine for Epstein-Barr virus-associated cancers”, as far as I know there is no approved mRNA vaccine against EBV.
Response: We modified the sentence accordingly. Page 19 lanes 30-31.
Line 386” Recent advancements in mRNA technology and the COVID emergency have brought the eyes to mRNA vaccines in the last years, and mRNA vaccines have emerged as a promising therapeutic strategy against cancer.”, no mRNA vaccines against cancer have been tested and optimised before 2020 and thanks to this the COVID vaccines could be developed. The first mRNA vaccine clinical study published was an anticancer (melanoma) vaccine by Weide et al 2008. Line 398 “However, none of them had managed to advance into phase III.”, no there is one in phase III as mentioned next line! Some important publications are missing.
Response: Agree. Page 20 lanes 13-16.
Reviewer 3 Report
Comments and Suggestions for Authors
The manuscript "The Bright Future of mRNA as a Therapeutic Molecule" presents a thorough review of mRNA technology, particularly focusing on its applications in infectious diseases and cancer. While the work is scientifically sound and well-organized, there is a stronger emphasis on mRNA’s role in COVID-19 vaccines, with less discussion on its emerging potential in cancer immunotherapy. Expanding the manuscript’s coverage on personalized mRNA cancer vaccines, recent clinical trials, and advanced delivery systems specifically for oncology would enhance its depth. Additionally, minor adjustments in language and structure would further improve readability. Overall, the manuscript is a valuable contribution, but greater attention to cancer-focused applications and the latest developments would significantly increase its impact.
Below are my scientific and technical comments to improve the quality of the publication:
1. The historical context provided on mRNA vaccines, from Wolff's study to the COVID-19 pandemic, is useful. However, recent advancements in cancer immunotherapy should be introduced earlier. Consider adding references to recent clinical trials or FDA approvals related to mRNA-based cancer vaccines, which are currently discussed much later in the manuscript. Including a brief explanation of mRNA's role in cancer immunotherapy early on would help set the stage for its applications beyond infectious diseases.
2. The focus in Section-3 is largely on COVID-19, while recent developments in oncology-focused mRNA vaccines are underemphasized. Expand this section to include more current advancements in mRNA cancer vaccines. Address challenges specific to cancer, such as the tumor microenvironment and immune evasion, and how mRNA vaccines aim to overcome these hurdles.
3. Recent studies on poly(A) modifications for cancer-specific applications are briefly mentioned but lack cancer-specific references. Include recent studies in this area, especially regarding poly(A) tail optimization for mRNA stability in cancer therapy.
4. The Section-7 is crucial but could benefit from more depth on personalized cancer vaccines. While mRNA-4157 is mentioned, a more detailed discussion of personalized neoantigen-based vaccines and their mechanisms would strengthen the manuscript. Consider adding clinical trial results to provide concrete examples of recent success in mRNA cancer immunotherapy.
5. The figure-3 illustrating the mechanisms of mRNA cancer vaccines is well done. However, expanding the text to explain how tumor-specific immune responses differ from those in infectious diseases would be beneficial. A more detailed discussion of the roles of CD4+ T cells and B cells in forming memory responses against tumor cells, preventing relapse, would improve this section.
6. The discussion of LNPs is thorough, but references to newer LNP delivery systems designed specifically for oncology are missing. Include more recent studies on targeted delivery in cancer using modified LNPs, such as antibody-conjugated LNPs. Additionally, a comparative table summarizing different nanoparticle delivery platforms (LNPs, viral vectors, peptides) for various diseases could enhance clarity.
7. The patent section is well-presented, but more context on how these patents have influenced real-world applications, particularly in oncology, would be valuable. Highlight the patents directly advancing mRNA cancer therapies in Table 2.
8. The conclusion could be more forward-looking by addressing remaining challenges for mRNA cancer therapeutics (e.g., organ-specific delivery, resistance mechanisms in tumors) and discussing future directions, such as combination therapies with mRNA vaccines and checkpoint inhibitors or adoptive cell therapies.
9. While references are appropriate, more up-to-date literature, especially from 2023-2024, is needed. This is particularly important for mRNA cancer vaccine clinical trials and emerging delivery systems like LNPs. Reference recent developments in LNPs for cancer therapy beyond COVID-19 applications.
10. The recognition of Karikó and Weissman’s contributions is well addressed, but it’s essential to highlight the applications of their pseudouridine work in cancer vaccines.
11. Overall, the manuscript covers a broad scope, but balancing the discussion between cancer and infectious disease applications is crucial. Consider reorganizing the manuscript to give more prominence to oncology, especially in the abstract, introduction, and early sections. Shortening the sections focused on infectious diseases, or including more recent cancer-related examples, would help ensure the manuscript remains relevant to the target audience.
Author Response
Reviewer #3
The manuscript "The Bright Future of mRNA as a Therapeutic Molecule" presents a thorough review of mRNA technology, particularly focusing on its applications in infectious diseases and cancer. While the work is scientifically sound and well-organized, there is a stronger emphasis on mRNA’s role in COVID-19 vaccines, with less discussion on its emerging potential in cancer immunotherapy. Expanding the manuscript’s coverage on personalized mRNA cancer vaccines, recent clinical trials, and advanced delivery systems specifically for oncology would enhance its depth. Additionally, minor adjustments in language and structure would further improve readability. Overall, the manuscript is a valuable contribution, but greater attention to cancer-focused applications and the latest developments would significantly increase its impact.
Below are my scientific and technical comments to improve the quality of the publication:
- The historical context provided on mRNA vaccines, from Wolff's study to the COVID-19 pandemic, is useful. However, recent advancements in cancer immunotherapy should be introduced earlier. Consider adding references to recent clinical trials or FDA approvals related to mRNA-based cancer vaccines, which are currently discussed much later in the manuscript. Including a brief explanation of mRNA's role in cancer immunotherapy early on would help set the stage for its applications beyond infectious diseases.
Response: Agree. Page 3 lanes 1-12, and page 32 lanes 84-100.
- The focus in Section-3 is largely on COVID-19, while recent developments in oncology-focused mRNA vaccines are underemphasized. Expand this section to include more current advancements in mRNA cancer vaccines. Address challenges specific to cancer, such as the tumor microenvironment and immune evasion, and how mRNA vaccines aim to overcome these hurdles.
Response: Page 34 lanes 128-147 and page 21 lanes 1-5. We also included the additional Fig. 4.
- Recent studies on poly(A) modifications for cancer-specific applications are briefly mentioned but lack cancer-specific references. Include recent studies in this area, especially regarding poly(A) tail optimization for mRNA stability in cancer therapy.
Response: No time for this. The deadline was too close to address all three reviewers´ comments.
- The Section-7 is crucial but could benefit from more depth on personalized cancer vaccines. While mRNA-4157 is mentioned, a more detailed discussion of personalized neoantigen-based vaccines and their mechanisms would strengthen the manuscript. Consider adding clinical trial results to provide concrete examples of recent success in mRNA cancer immunotherapy.
Response: Page 21 lanes 1-5.
- The figure-3 illustrating the mechanisms of mRNA cancer vaccines is well done. However, expanding the text to explain how tumor-specific immune responses differ from those in infectious diseases would be beneficial. A more detailed discussion of the roles of CD4+ T cells and B cells in forming memory responses against tumor cells, preventing relapse, would improve this section.
Response: Page 30 lanes 25-33, page 31 lanes 34-51, and page 32 lanes 69-81 and lanes 84-100.
- The discussion of LNPs is thorough, but references to newer LNP delivery systems designed specifically for oncology are missing. Include more recent studies on targeted delivery in cancer using modified LNPs, such as antibody-conjugated LNPs. Additionally, a comparative table summarizing different nanoparticle delivery platforms (LNPs, viral vectors, peptides) for various diseases could enhance clarity.
Response: Page 14 complete and page 15 lanes 1-16.
- The patent section is well-presented, but more context on how these patents have influenced real-world applications, particularly in oncology, would be valuable. Highlight the patents directly advancing mRNA cancer therapies in Table 2.
Response: Very interesting point, but no time for this. The deadline was too close to address all three reviewers´ comments.
- The conclusion could be more forward-looking by addressing remaining challenges for mRNA cancer therapeutics (e.g., organ-specific delivery, resistance mechanisms in tumors) and discussing future directions, such as combination therapies with mRNA vaccines and checkpoint inhibitors or adoptive cell therapies.
Response: page 35 lanes 159-170.
- While references are appropriate, more up-to-date literature, especially from 2023-2024, is needed. This is particularly important for mRNA cancer vaccine clinical trials and emerging delivery systems like LNPs. Reference recent developments in LNPs for cancer therapy beyond COVID-19 applications.
Response: Done. page 6 lanes
- The recognition of Karikó and Weissman’s contributions is well addressed, but it’s essential to highlight the applications of their pseudouridine work in cancer vaccines.
Response: Agree. Page 6 lanes 21-30.
- Overall, the manuscript covers a broad scope, but balancing the discussion between cancer and infectious disease applications is crucial. Consider reorganizing the manuscript to give more prominence to oncology, especially in the abstract, introduction, and early sections. Shortening the sections focused on infectious diseases, or including more recent cancer-related examples, would help ensure the manuscript remains relevant to the target audience.
Response: Agree. Done.
Round 2
Reviewer 2 Report
Comments and Suggestions for Authors
The authors incorporated the corrections I proposed but still made mistakes. For example " Later, Conry et al. induced humoral immune response upon the inoculation of an mRNA encoding the human carcinoembryonic antigen in mice": no, this article did not show induction of a humoral immune response on its own, they showed higher antibody response after tumor challenge in mice having received mRNA injections. "due to the suppression of RNA recognition by TLRs", no modifications do not suppress RNA recognition by TLR, they avoid it. " however, they are usually less efficient than those using modified nucleotides because are prone to quick degradation and there may be mild, transient side effects due to stronger immune activation. ", no, non-modified mRNA vaccines are as good or even better (Arcturus: 5ug per injection and high and prolonged antibody response) than the modified mRNA vaccines.
Author Response
Responses to Reviewer #2 comments round2:
Once again, we thank Reviewer #2 for the new critical comments and suggestions.
Here are the point-by-point responses to the reviewers´ comments and the modifications to the manuscript accordingly, which have been highlighted in yellow in the new version (Version 4) .
Comment: "Later, Conry et al. induced humoral immune response upon the inoculation of an mRNA encoding the human carcinoembryonic antigen in mice": no, this article did not show induction of a humoral immune response on its own, they showed higher antibody response after tumor challenge in mice having received mRNA injections.
Response: We agree. We didn’t give enough information to lead to the correct interpretation about Conry’s work. We corrected those lines. Page 5, lanes 10-14 of the new version.
Comment: "due to the suppression of RNA recognition by TLRs", no modifications do not suppress RNA recognition by TLR, they avoid it
Response: We agree, and we have changed it accordingly. Page 7, lanes 5-18 of the new version
Comments: "however, they are usually less efficient than those using modified nucleotides because are prone to quick degradation and there may be mild, transient side effects due to stronger immune activation. "
Response: We agree and have corrected the wrong information. Page 7, lanes 5-18 of the new version.